# Study on modular smoke extraction with solid screen in urban road tunnel fires

**Xiaotao Zhang** [iD]**[1]\*, Kaihua Lu[2], Yushi Lu[2]**

**1** School of Civil Engineering and Architecture, Suqian university, Suqian, Jiangsu, China, **2** Faculty of Engineering, China University of Geosciences, Wuhan, Hubei, China

\* zhangxt@squ.edu.cn

## Abstract

A series of numerical simulations were conducted to evaluate and optimize a modular smoke extraction system integrated with smoke screens for urban road tunnel fires. The study aimed to identify key design parameters and propose optimal strategies for tunnel-wide smoke management. The results show that smoke screen height is a critical factor, with a threshold of 1.5 m (25% of tunnel height) significantly enhancing extraction efficiency and reducing high-temperature zones, while vent-screen distance has minimal impact. Dividing the tunnel into five modular zones achieves the optimal balance between hazard control and system cost. Furthermore, an unbalanced airflow distribution strategy—allocating 35% of the total exhaust volume (180 m³/s) to proximate vents and 15% to distal vents—proves most effective in preventing plug-holing phenomenon and maximizing CO extraction efficiency. These findings provide specific design thresholds and actionable strategies for the optimization of transverse smoke extraction systems in urban road tunnels.

## Introduction

Urban tunnels are critical components of modern transportation infrastructure, facilitating the flow of vehicles and alleviating surface congestion. However, their enclosed nature makes them particularly vulnerable to fire hazards [1]. In such incidents, high temperatures and rapidly accumulating smoke severely impair visibility and hinder evacuation, posing substantial risks to human safety, structural integrity, and the environment [2,3]. A recent International Fire Academy study of tunnel fires in Switzerland, Germany, and Austria (2012–2023) recorded 439 incidents, approximately 75% of which occurred in urban tunnels. This highlights a pronounced and growing risk in urban tunnel type [4]. The data underscores the urgent need for enhanced fire prevention and smoke extraction measures, and emphasizes the crucial role of efficient smoke extraction systems in urban tunnels. Effective smoke control in urban tunnels is vital not only for protecting lives during fire emergencies but also for minimizing environmental contamination by preventing the spread of toxic smoke beyond tunnel

**Data availability statement:** All relevant data are within the manuscript and its Supporting information files.

**Funding:** The works described in this paper were supported by the following grants: the Suqian Science and Technology Program (Grant No. K202434) to X.Z.; the Suqian Social Science Research Project (Grant No. 25SYT-11) to X.Z.; the National Natural Science Foundation of China (Grant No. 52376133) to X.Z.; and the Jiangsu Social Science Foundation Project (Grant No. 23GLD009) to X.Z. The funders had no role in study design, data collection and analysis, decision to publish, or preparation of the manuscript.

**Competing interests:** The authors have declared that no competing interests exist.

boundaries [5]. As cities expand their underground transportation networks, the need for robust, adaptable smoke extraction methods becomes more pressing.

Traditional mechanical smoke extraction in urban tunnels is primarily achieved through two systems: longitudinal and transverse [6,7]. Longitudinal systems use high-powered jet fans to direct smoke along the tunnel length toward the exits. This straightforward approach makes them a cost-effective choice in tunnel fire safety design [8]. In contrast, transverse systems employ a network of ducts and strategically placed exhaust vents along the tunnel ceiling or walls. This design enables more localized smoke extraction, aiming to reduce smoke concentration directly at the fire source [9,10]. From a safety perspective, transverse smoke extraction offers significant advantages over longitudinal systems. It can localize smoke removal, reducing smoke spread and accumulation across the length of the tunnel. This results in better control of the smoke layer, a lower risk of smoke infiltrating evacuation routes, and ultimately, enhanced safety and visibility for both occupants and first responders. Furthermore, this approach minimizes the potential for smoke re-circulation, thereby creating a more effective and safer evacuation environment [11,12].

Current research on transverse smoke extraction primarily focuses on optimizing exhaust vent placement, airflow rates, and the coordination of multiple extraction points to maximize smoke extraction efficiency [13]. Li et al. [11] investigated smoke control in ultra-wide tunnels under different exhaust patterns and longitudinal air supply volumes. Their results demonstrated that a top exhaust pattern with a 50% air supply ratio significantly enhanced control performance. Han et al. [14] studied the effect of transverse ventilation on smoke propagation in urban tunnel fires. The study, based on 14 full-scale tests in Chongqing, China, explored the impact of fire location, air supply, and smoke extraction on induced air velocity and ceiling temperature. Xu et al. [15] evaluated smoke exhaust performance under a lateral centralized mode, employing both theoretical models and FDS simulations. Their results indicated that increasing the exhaust volume improves the efficiency of both heat and smoke removal. The optimal performance was observed at a single-sided exhaust volume of 180 $m^3$/s, achieving heat and smoke exhaust efficiencies of 54.2% and 47.3%, respectively.

Recent studies have increasingly explored the use of physical barriers—such as solid screens or smoke curtains—to better control smoke movement within tunnels. Murakami et al. [16] used numerical simulations to investigate Water Screens (WS) for fire compartmentalization. Their work demonstrated that WS effectively improve smoke control by enhancing compartmentalization and thereby reducing the spread of heat and smoke. Halawa [17] examined the impact of installing solid curtains near extraction vents in road tunnels. The results indicated that optimal smoke control is achieved when the curtain is placed at a distance of 90% of the tunnel height and its own height is between 16% and 30% of the tunnel height. Chaabat et al. [18] focused on damper shape and position for smoke confinement under transverse ventilation. They found that full-width dampers performed best, minimizing backflow and maintaining stable stratification, whereas ceiling-mounted square dampers disrupted the smoke layer. In a numerical study of transverse extraction in immersed tunnels, Zhang et al. [19] introduced a novel inclined smoke screen. Their results showed

that this design enhances extraction efficiency, particularly for smaller fires, with optimal performance occurring when the screen is inclined at an angle between 30° and 75°. Another study by Zhang et al. [20] investigated solid screens in urban road tunnels equipped with vertical shafts. They reported increased extraction efficiency when the screen was placed 0.3–1.5 m from the shaft vent and had a height of 0.9–1.5 m.

To provide a clearer presentation of the research context, Table 1 summarizes the literature on mechanical smoke control in tunnel fires reviewed above.

Based on the evidence from these studies, existing research has established smoke screens as an effective enhancement to transverse smoke extraction systems in tunnel fires. However, the focus of research on smoke screens has remained predominantly on localized smoke control in isolated or small-scale sections, often neglecting the challenges of coordinating extraction across the entire tunnel network. Consequently, strategies for comprehensive smoke management in full-scale urban tunnel fire scenarios have not been sufficiently addressed.

To address this need, this study undertakes a systematic numerical investigation using Computational Fluid Dynamics (CFD) to evaluate a novel modular smoke extraction system with solid screens in a full-scale urban road tunnel. The primary contributions are threefold: (1) establishing a parametric framework to quantify the combined effects of key design variables on global extraction efficiency; (2) identifying optimal design thresholds that balance performance with practical cost; and (3) proposing specific engineering guidelines, including an unbalanced airflow strategy, for system optimization. These outcomes provide direct, actionable insights for the design of advanced transverse ventilation systems in urban tunnels.

## Research framework

To systematically address the research objective, a parametric study integrated with Computational Fluid Dynamics (CFD) simulations was conducted. The overall workflow and logical structure of this investigation are summarized in Fig 1. The process begins with defining the research objective, followed by the establishment of a detailed numerical model

**Table 1. Summary of key literature on mechanical smoke control in tunnel fires.**

| Ref. | Method | Core Research Focus |
|---|---|---|
| [1] | Review | Overview of tunnel fire safety design, including smoke control. |
| [2] | Review | Foundational small-scale duct experiments on fire spread and ventilation. |
| [3] | Review | Fire hazards and mitigation strategies in transportation infrastructures. |
| [4] | Statistical Analysis | Analysis of tunnel fire incident statistics (2012–2023). |
| [5] | Experiment & Theory | Smoke control strategy and design criterion for "complete smoke extraction". |
| [6] | Review | Comparison of longitudinal and transverse ventilation systems for different tunnel types. |
| [7] | Experiment | Comparison of longitudinal vs. transverse ventilation effectiveness in a UTLT. |
| [8] | Experiment/Simulation | Smoke stratification length under longitudinal ventilation. |
| [9] | Theory/Simulation | Quantitative evaluation and optimization of exhaust under lateral centralized mode. |
| [10] | Simulation/Experiment | Influence of vent number/layout on full transverse exhaust performance. |
| [11] | Simulation | Smoke control with different exhaust patterns and air supply in ultra-wide tunnels. |
| [12] | Simulation | Effect of lateral extraction on transverse temperature distribution under ceiling. |
| [13] | Simulation | Mechanical smoke extraction efficiency of multiple lateral vents in an immersed tunnel. |
| [14] | Experiment | Impact of transverse ventilation on smoke spread in an urban tunnel. |
| [15] | Theory/Simulation | Heat and smoke exhaust performance under lateral centralized mode. |
| [16] | Simulation | Compartmentalization and smoke control using Water Screens (WS). |
| [17] | Simulation | Optimal placement and height of solid curtains near exhaust vents. |
| [18] | Experiment | Damper shape and position for smoke confinement under transverse ventilation. |
| [19] | Simulation | A novel inclined smoke screen for lateral extraction in an immersed tunnel. |
| [20] | Simulation | Solid screen enhancement for shaft extraction in urban road tunnels. |

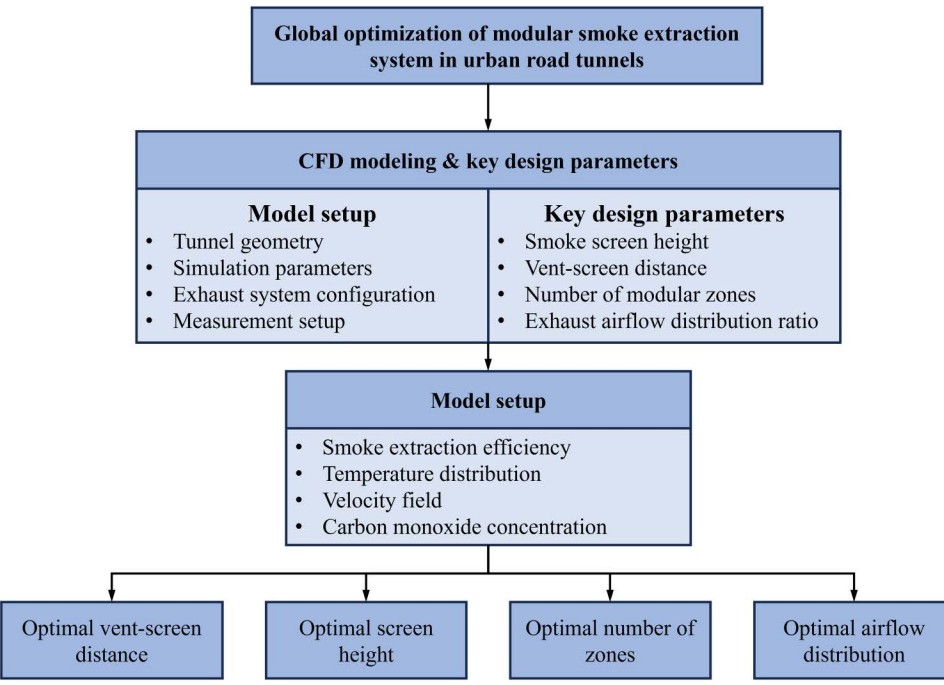

**Fig 1. Flowchart of the research framework.**

encompassing both the tunnel geometry and the modular exhaust system configuration. Key design parameters were then varied within this model. System performance was evaluated against a set of predefined metrics, ultimately leading to the identification of optimal design parameters for practical application.

## Methods

### Basis of model construction

The Fire Dynamics Simulator (FDS), developed by the National Institute of Standards and Technology (NIST), is a computational tool widely validated for tunnel fire research. In this study, FDS version 6.8.0 is employed to simulate various fire scenarios. This version introduces an enhanced combustion model that improves the accuracy of chemical reactions, supporting a broader range of fuel types and incomplete combustion cases. These advancements allow for more reliable predictions of heat release rate (HRR) and flame spread in multi-material and multi-source fire scenarios. Smoke flow is modeled using an improved Large Eddy Simulation (LES) framework, which enhances the simulation of smoke movement in complex geometries. Additionally, the accuracy of smoke control system simulations has been refined, optimizing predictions for smoke stratification, backflow, and the performance of both mechanical and natural ventilation systems. These updates make FDS 6.8.0 particularly suited for complex fire and smoke flow scenarios, which offer significant improvements in fire modeling for tunnels, underground structures, and other intricate environments [21].

### Model assumptions and limitations

To establish a controlled baseline for evaluating the core performance of the modular extraction system, the following simplifying assumptions were adopted in the numerical model:

(1)  Both portals were modeled as open boundaries, neglecting potential semi-enclosed conditions or additional ventilation shafts.

(2) The tunnel was assumed to be straight and horizontal, without curvature or slope.

(3) External influences such as longitudinal wind, natural draft, and vehicle-induced flows were excluded.

(4) A steady-state n-heptane pool fire with a constant HRR was used, omitting the transient growth phase.

## Model construction

A typical full-scale urban road tunnel was constructed for simulation. The tunnel has a rectangular cross-sectional dimension of 10.0 m × 6.0 m (width × height) and a total length of 400 m, as illustrated in Fig 2. This cross-section represents a standard bidirectional four-lane urban tunnel in China, complying with the design specification for highway tunnels in China [22], and has been widely adopted in prior tunnel fire research [15,23,24]. The tunnel walls and smoke screens were defined as "INERT," representing non-reactive surfaces. Both ends of the tunnel were modeled as open boundaries, simulating the connection to an outdoor environment. Ambient conditions were set with a temperature of 293 K and atmospheric pressure of 101.325 kPa to reflect standard environmental conditions. The simulation runtime was determined as 600 seconds, based on the time required for the smoke temperature and CO concentration to reach a steady state.

A fire source using n-heptane (representing gasoline) as fuel was located at the center of the tunnel floor (x = 0 m, y = 5 m, z = 0 m), with a burning surface area of 4.0 m × 2.0 m. The heat release rate (HRR) was set to 20 MW, simulating a truck fire scenario in a road tunnel [15]. The "simple chemistry" combustion model was used [20]. Based on previous large-scale gasoline pool fire tests, the CO yield and soot yield were set to 0.05 each [25].

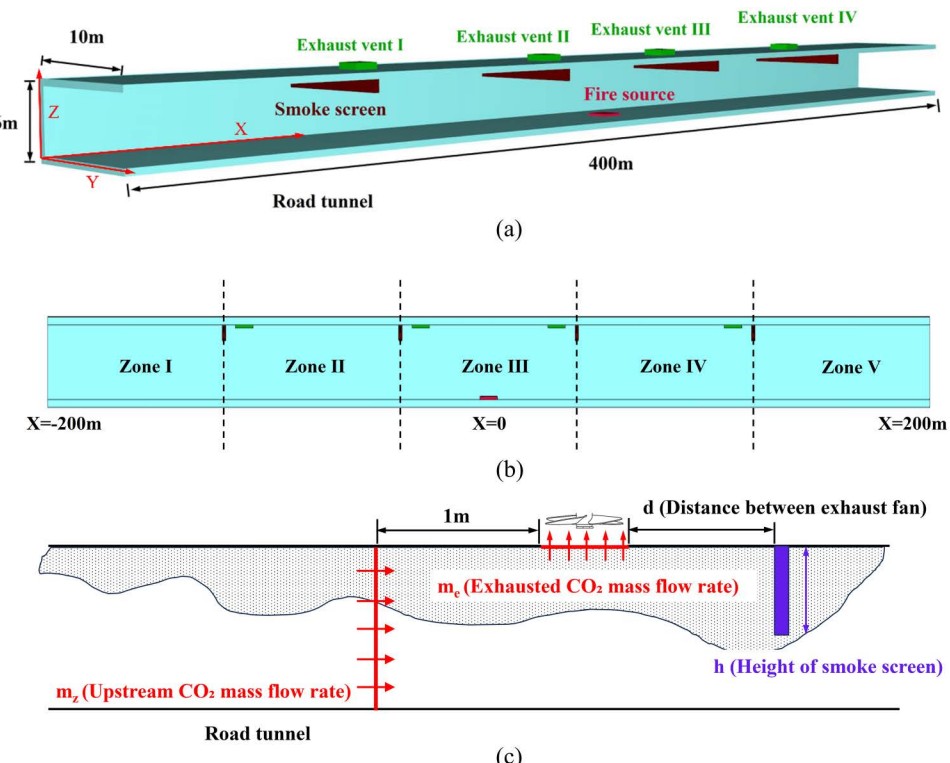

**Fig 2. Schematic diagram of model road tunnel and measurement of $CO_2$ mass flow rate in simulations. (a)** model of road tunnel; **(b)** Zoning configuration of the tunnel; **(c)** Schematic of $CO_2$ mass flow rate measurement.

Mesh size plays a critical role in determining the simulation's accuracy and must be carefully considered. Generally, smaller mesh sizes yield more precise computational results but significantly increase computational effort and cost. Previous studies [26,27] indicate that when the mesh size is less than 0.1 D*, the results are acceptable. The D* can be determined by:

$$Q^* = \left[ \frac{Q}{\rho_0 c_p T_0 \sqrt{g}} \right]^{2/5}$$

(1)

The characteristic diameter (D*) was calculated to be 3.54 m for a fire with a heat release rate of 20 MW, resulting in 0.1 D* being approximately 0.35 m. To verify that this mesh resolution yields reliable results, a sensitivity analysis was conducted. Three non-uniform mesh cases were designed and compared:

- Case M1:0.125 m cells in the fire region (−25–25 m) and 0.25 m elsewhere.

- Case M2: 0.25 m cells in the fire region (−25–25 m) and 0.5 m elsewhere.

- Case M3: 0.5 m cells in the fire region (−25–25 m) and 1.0 m elsewhere.

A temperature monitor was positioned below the center of exhaust vent III (Z = 5.0 m). The resulting temperature-time profiles under the three mesh cases are compared in Fig 3.

The mesh sensitivity results show that Case M2 and Case M1 produce nearly identical temperature trends, whereas Case M3 exhibits larger fluctuations. Given that Case M1 incurs a computational cost over ten times higher than Case M2, the 0.25/0.5 m non-uniform mesh (Case M2) was selected as the optimal compromise between accuracy and efficiency.

Accurate subgrid-scale parameterization is crucial for the reliability of Large Eddy Simulation (LES) in modeling buoyancy-driven fire flows. Three parameters are recognized as particularly influential: the Smagorinsky constant (Cs), the turbulent Prandtl number (Pr), and the turbulent Schmidt number (Sc). For strong buoyancy-driven flows typical of

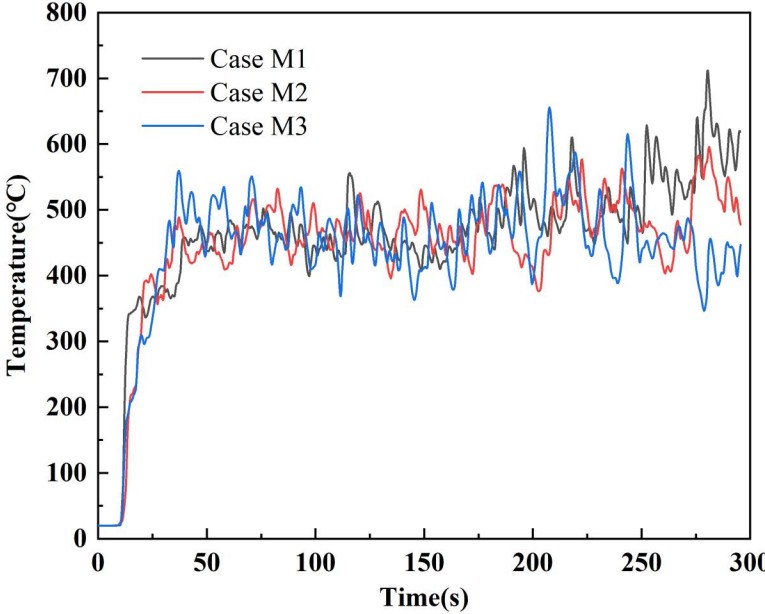

**Fig 3. Temperature-time profiles from the mesh sensitivity analysis.**

compartment fires, Zhang's studies have established that a value of Cs = 0.18 yields accurate turbulence statistics [28]. Furthermore, the values Pr = 0.5 and Sc = 0.5 have been extensively validated as the default and physically consistent settings for fire and smoke simulations within the FDS framework [21]. Following these established guidelines, the present study adopts the validated parameter set: Cs = 0.18, Pr = 0.5, and Sc = 0.5, to ensure a physically sound turbulence closure.

To complement the parameter justification, the simulation setup was validated by replicating the experimental study of Luo et al. [29]. As shown in Fig 4, the simulated temperature profiles show good agreement with the measured data, confirming that the adopted model and parameters reliably capture the key physics of smoke flow in a confined fire scenario.

Movable smoke screens partition the tunnel into zones (e.g., five zones in Fig 1a). Due to space constraints in road tunnels (typically 5–6 m height with >4 m clearance for fire trucks), these ceiling-mounted screens deploy automatically via fire detectors [30,31]. We tested five screen heights (0–2 m in 0.5 m increments) to optimize performance.

The exhaust vents (each 4.0 m × 2.0 m) are installed on the 6 m-high tunnel ceiling, with one vent per zone. Vents are labeled sequentially from the tunnel's left end (I, II, III, IV, etc.). Zhang et al. reported optimal natural smoke extraction at 0.3–1.5 m vent-screen distance in vertical shafts [20]. This study extends the principle to mechanical systems by testing 1.0–3.0 m vent-screen distances. For a 20 MW HRR, Xu et al. [15] recommended a total exhaust airflow of 180 m³/s to optimize heat and smoke extraction. Accordingly, 180 m³/s is initially distributed equally among vents for further ratio optimization.

Several testing devices were deployed to monitor key parameters, including CO concentration, $CO_2$ mass flux, smoke temperature, and smoke velocity field, as follows:

- **CO concentration**: Following evacuation safety standards, detectors were installed at 2 m height (above ground) along the tunnel centerline, spaced 1 m apart to record real-time CO variations.

- **$CO_2$ mass flux**: The "Statistics" function in FDS was employed to record the $CO_2$ mass flow rate both upstream of each exhaust vent and at the vent outlet (Fig 1c).

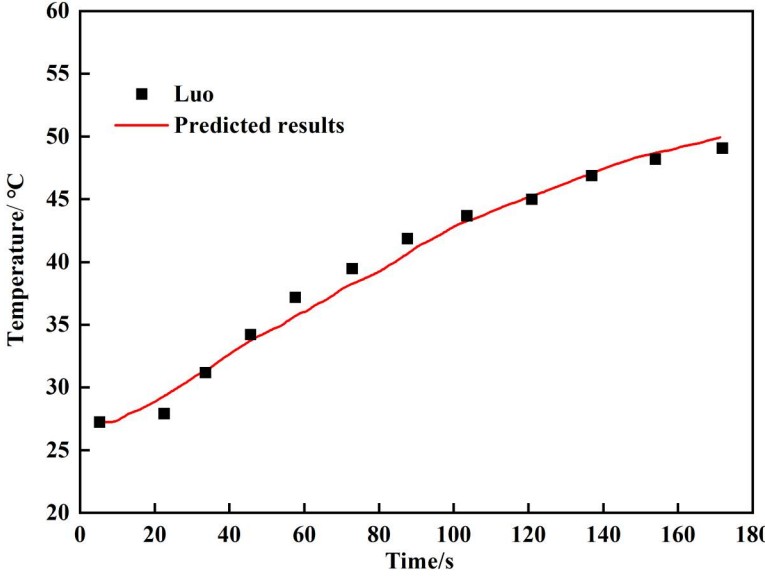

**Fig 4. Model validation against experimental temperature data.**

- **Smoke temperature**: A temperature slice was positioned at the tunnel's cross-sectional center to capture the temperature distribution of the smoke layer.

- **Smoke velocity**: A velocity vector slice was also configured at the tunnel's cross-sectional center to illustrate the velocity field characteristics of smoke flowing into the exhaust vent.

## Results and discussion

### Effect of screen height and vent-screen distance on smoke extraction efficiency

This section analyzes the impact of smoke screen height ($h$) and vent-screen distance ($d$) on smoke control efficacy. The tunnel is divided into five zones by four smoke screens and four exhaust vents, with a total exhaust airflow of 180 $m^3/s$. Numerical simulation scenarios are detailed in Table 2. We systematically evaluate how $h$ and $d$ variations affect extraction efficiency and overall smoke management performance.

Smoke exhaust efficiency is a key parameter for evaluating the performance of smoke extraction systems [32]. It is defined as the ratio of the total smoke volume flow entering each zone to the smoke volume extracted through the exhaust vent in that zone. Since $CO_2$ features a stable yield for the specified heptane fuel and is straightforward to monitor in FDS, the mass flux of $CO_2$ is employed as a robust proxy for the convective smoke mass flow. This approach is common practice in the comparative evaluation of mechanical smoke extraction systems for tunnels [11,20]. The smoke extraction efficiency can be determined using the following equation:

$$\eta = \frac{m_e}{m_z} \times 100\%$$

(2)

Here, $\eta$ represents the exhaust efficiency of each exhaust vent (%), $m_e$ denotes the mass flow rate of $CO_2$ exhausted through the vent in each zone (kg/s), and $m_z$ denotes the $CO_2$ mass flow rate within the tunnel before reaching the exhaust vent. Fig 5 illustrates the calculated smoke extraction efficiencies of each exhaust vent under varying smoke screen height and vent-screen distance.

As shown in Fig 5, the exhaust efficiency of vents II and III is notably lower (minimum: 17.5%) compared to vents I and IV (maximum: 38.8%). This discrepancy stems from the proximity of vents II/III to the fire source, where strong buoyancy-driven horizontal smoke velocity reduces the relative suction force, hindering smoke capture. In contrast, vents I/IV, positioned farther from the fire, benefit from attenuated buoyancy and wall friction-induced velocity decay, enabling more effective smoke extraction.

Fig 5 further shows that increasing the vent-screen distance (d) from 1 m to 3 m has a negligible impact on smoke exhaust efficiency, indicating that vent-screen distance is a secondary factor within this range. This finding contrasts with that of Zhang et al. [20] for a natural ventilation system. In their study, exhaust efficiency was highly sensitive to shaft-screen distance over a short range. This discrepancy likely stems from the different driving mechanisms involved: natural

**Table 2. Parametric study cases for smoke screen configuration.**

| Case No. | $h$ (m) | $d$(m) | Exhaust volume ($m^3/s$) | | | |
|---|---|---|---|---|---|---|
| | | | I | II | III | IV |
| H1~H3 | 0 | 1,2,3 | 45 | 45 | 45 | 45 |
| H4~H6 | 0.5 | | | | | |
| H7~H9 | 1.0 | | | | | |
| H10~H12 | 1.5 | | | | | |
| H13~H15 | 2.0 | | | | | |

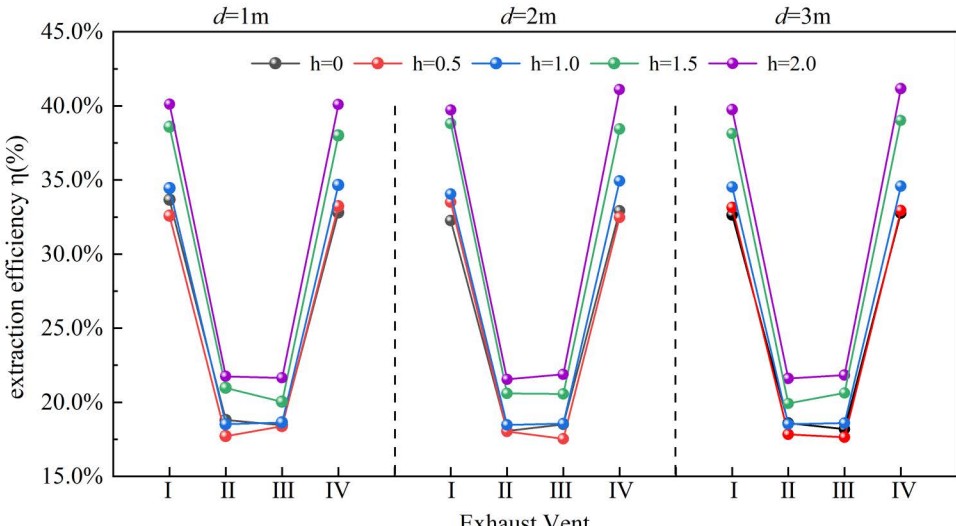

**Fig 5. Smoke extraction efficiency under varying smoke screen heights and vent-screen distances.**

ventilation relies primarily on thermal buoyancy, where even minor distance variations can alter the natural convergence path of hot smoke toward the shaft. In contrast, the mechanical transverse exhaust system employed here utilizes fan-induced forced pressure differences, which create an effective suction field across a wider region behind the screen (1–3 m). Consequently, capture efficiency is less sensitive to precise vent-screen spacing within this range.

In contrast to the vent-screen distance, the smoke screen height (h) plays a critical role in determining exhaust efficiency. When h = 0.5 m, the efficiency remains almost unchanged compared to the no-screen scenario. A slight improvement is observed at h = 1 m, while a marked efficiency surge occurs once h ≥ 1.5 m. This threshold behavior stems from a fundamental alteration in the smoke flow pattern induced by the screen: a lower screen (h ≤ 1.0 m) provides an insufficient effective blocking area to cause flow separation of the main smoke stream. Consequently, the smoke is only slightly perturbed and bypasses the screen with sustained high horizontal velocity. This flow pattern is directly observable in the horizontal velocity (denoted as U) distribution near the vent, as shown in Fig 6a–6c. When the screen height increases to 1.5 m (25% of the tunnel height), its effective blocking area becomes adequate to trigger significant flow separation, forming a stable recirculation vortex zone upstream of the screen. This zone corresponds to the extensive "negative velocity zone" observed in Fig 6d and 6e. Acting as a dynamic "smoke reservoir," it not only substantially dissipates the horizontal momentum of the smoke but also prolongs its residence time near the exhaust vent, thereby greatly enhancing the capture probability.

The critical screen height identified in this study (25%) aligns well with the effective screen-height ratio range (17%–29% of tunnel height) suggested by Zhang et al. [20] for natural ventilation shafts, confirming the applicability of this empirical guideline across different systems. Notably, the critical value here is closer to the upper limit of that range. This likely reflects an intrinsic difference between mechanical exhaust and natural ventilation systems: the active suction in mechanical systems exerts a stronger disturbance on the stability of the smoke layer, thus requiring a taller screen to maintain sufficient smoke layer thickness to counteract the enhanced airflow disturbance and ensure efficient smoke accumulation.

The temperature (denoted as T) distribution analysis in Fig 7 further corroborates these findings. Using 68°C (marked by red contours) as a critical threshold for irreversible human injury, it was observed that with screen heights <1.0 m, the hazardous zone remains comparable to no-screen conditions. When h ≥ 1.5 m, the high-temperature area shrinks

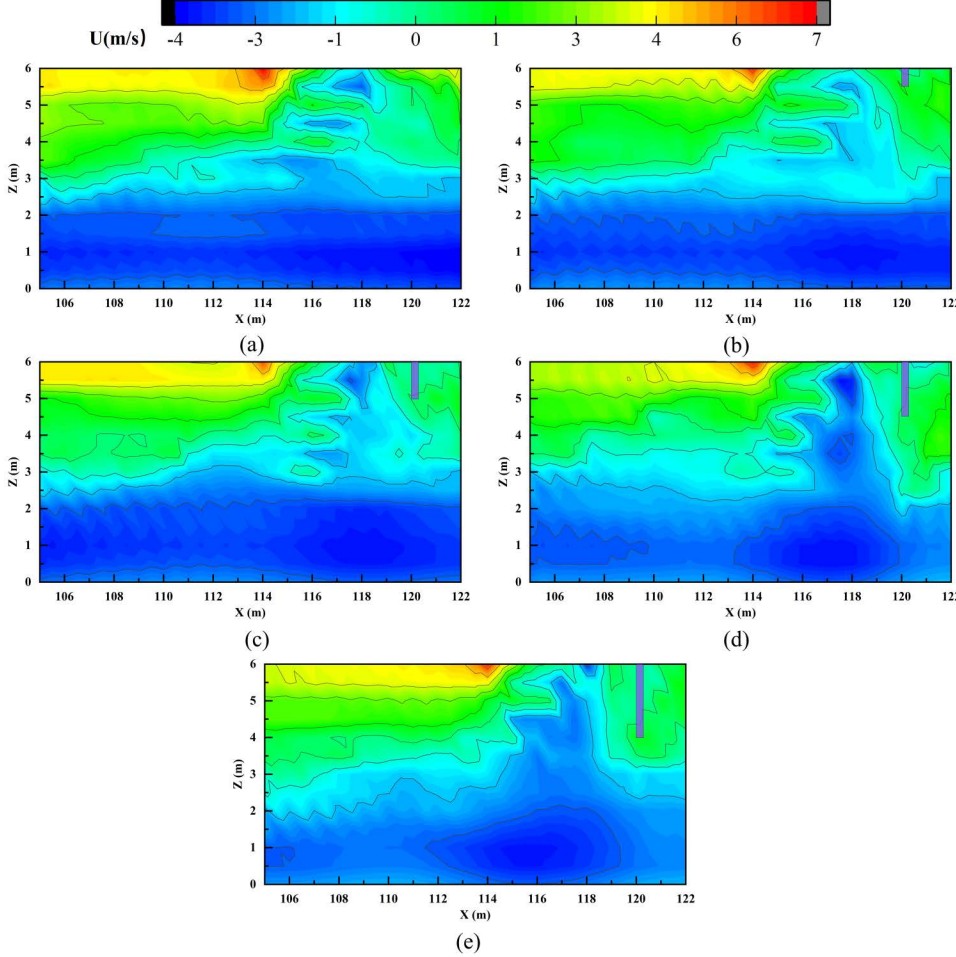

**Fig 6. Horizontal velocity distribution near exhaust vent with varying smoke screen heights: (a) h = 0m; (b) h = 0.5m; (c) h = 1.0m; (d) h = 1.5m; (e) h = 2.0m.**

significantly. Therefore, it can be concluded that a smoke screen height greater than 1.5 m effectively reduces the horizontal velocity of the smoke, improves exhaust efficiency, and reduces the area affected by high-temperature smoke.

### Effect of zone number on smoke extraction

The number of tunnel zones significantly impacts smoke control efficacy. While insufficient zones compromise containment, excessive zones incur unnecessary costs without proportional benefits. This study evaluates four zone configurations (Z3, Z5, Z7, Z9) with a fixed total airflow of 180 m³/s (evenly distributed), using optimized parameters from prior research: 1.5 m screen height and 2 m vent-screen spacing. Simulation details are provided in Table 3.

Carbon monoxide (CO), a highly toxic and rapidly diffusing gas [15,20], serves as a key indicator of smoke control efficacy. This study evaluates zoning scenarios by monitoring CO concentrations at a 2 m safety height. Fig 8 shows the CO concentration distribution along the tunnel's central axis, revealing peak values at the center and gradual decreases toward the ends. Based on established health thresholds [33]—200 ppm (causing headaches/fatigue) and 800 ppm (risk of death)—we classify:

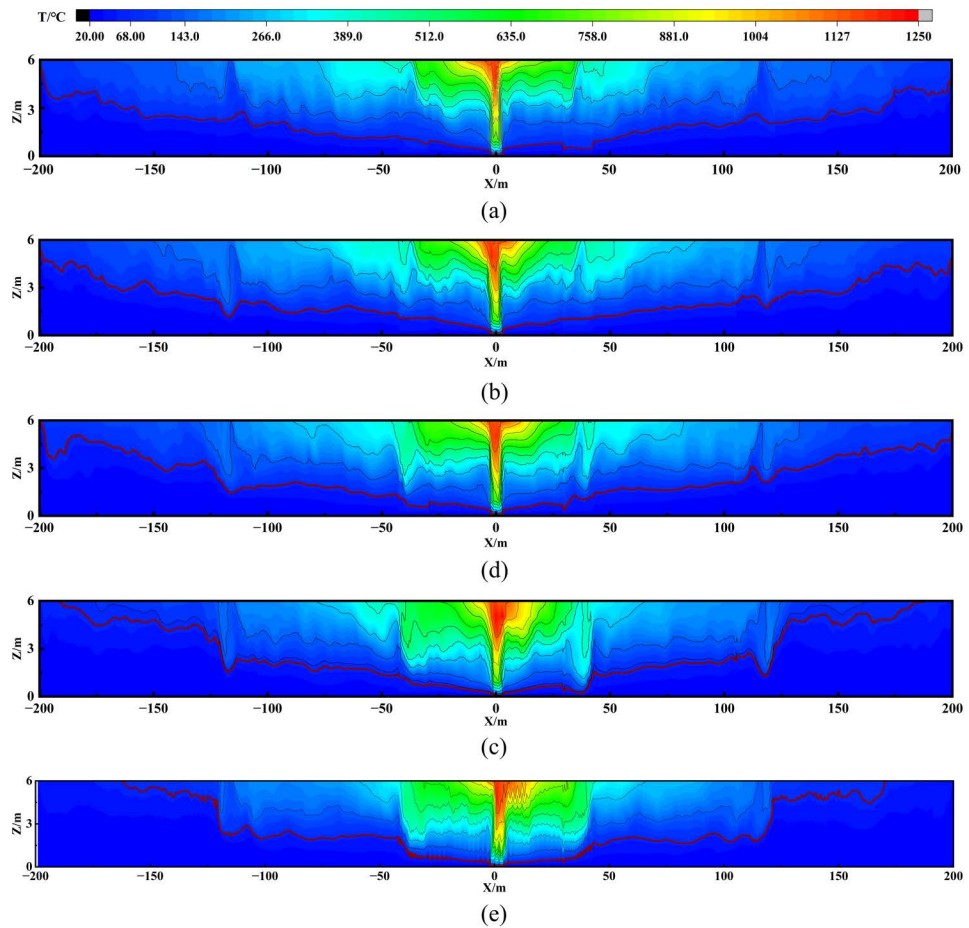

**Fig 7. Smoke temperature distribution in central surface with varying smoke screen heights: (a) h = 0m; (b) h = 0.5m; (c) h = 1.0m; (d) h = 1.5m; (e) h = 2.0m.**

**Table 3. Parametric study cases for zoning configuration.**

| Case No. | The number of tunnel zones | Exhaust volume per exhaust vent(m³/s) | h (m) | d (m) |
|---|---|---|---|---|
| Z3 | 3 | 90 | 1.5 | 2 |
| Z5 | 5 | 45 | | |
| Z7 | 7 | 30 | | |
| Z9 | 9 | 22.5 | | |

(1) Severe hazard zones: > 800 ppm;

(2) Moderate hazard zones: 200–800 ppm;

The spatial extents of these hazardous areas for each scenario are quantified in Fig 8b and 8c. The severe fire hazard zones span 132 m (Z3), 81 m (Z5), 114 m (Z7), and 130 m (Z9), while the moderate hazard zones cover 338 m (Z3), 249 m (Z5), 289 m (Z7), and 243 m (Z9). Smoke control efficacy for severe hazards ranks as Z5 > Z7 > Z9 > Z3, and for moderate hazards as Z9 > Z5 > Z7 > Z3. This difference stems from the combined effects of different zone configurations on

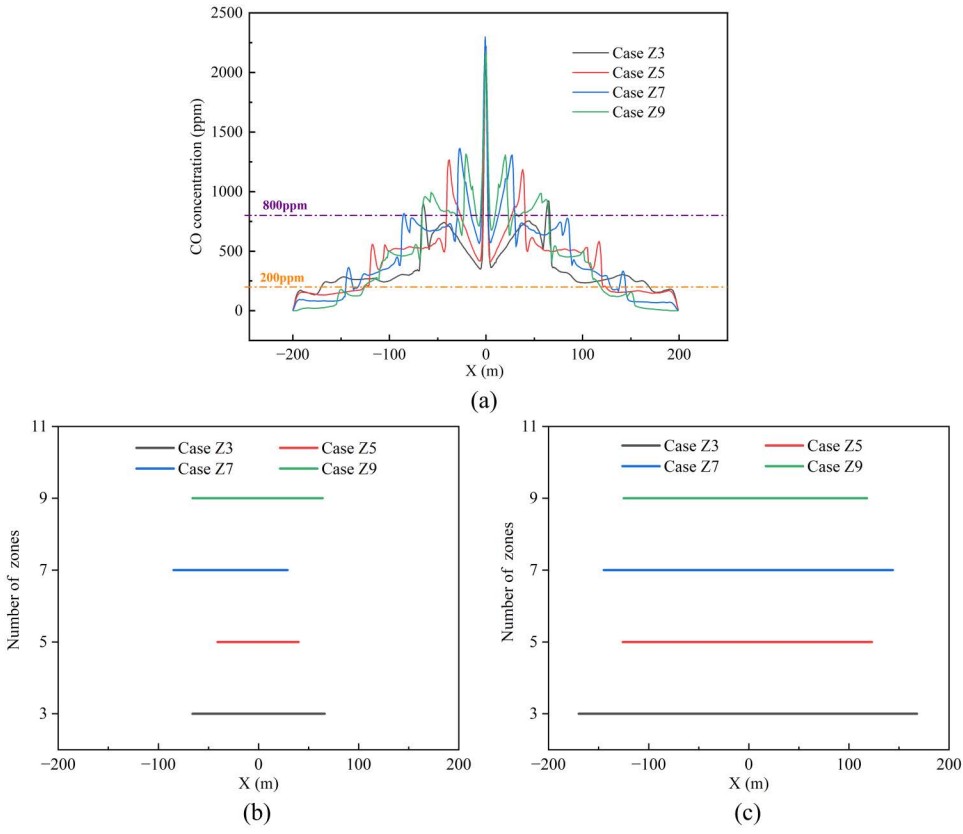

**Fig 8. Carbon monoxide concentration and hazard zone length ranges under different zone numbers: (a) CO concentration at safety height. (b)** Range of severe fire hazard zones. **(c)** Moderate fire hazard zones.

the "spatial interception density" of smoke and the "local suction intensity" of exhaust vents. In the Z3 configuration, an insufficient number of vents results in excessively large spacings between adjacent vents, creating an extensive "capture blind zone" downstream of the fire. Smoke spreads freely before reaching the first effective vent, leading to thickening of the smoke layer and consequently maximizing the longitudinal pollution range. Conversely, the Z9 configuration overly disperses the total airflow, resulting in significantly insufficient airflow at individual vents. Near the fire source, smoke possesses extremely high thermal buoyancy and horizontal momentum, while the suction force at vents in this region is insufficient to establish a dominant local pressure differential. Consequently, a large amount of high-concentration smoke is transported downstream, leading to a larger severe hazard zone. As smoke propagates downstream, its horizontal momentum decays due to wall friction and air entrainment. The diminished smoke inertia allows the suction effect of distant vents to become relatively more prominent. This coupling effect of "smoke momentum decay" and "relative enhancement of exhaust power" enables effective containment and extraction of downstream smoke, thereby reducing the extent of the moderate hazard zone. The Z5 configuration achieves the optimal balance: a sufficient number of partitions shortens the free diffusion distance of smoke, enabling early and effective interception, while the airflow allocated to each vent (especially those near the fire) generates sufficiently strong local suction to significantly suppress long-distance smoke propagation. As a result, the extents of both severe and moderate hazard zones are effectively controlled.

To move beyond qualitative comparison and provide a quantitative basis for selecting the optimal configuration, a benefit-to-cost evaluation framework was established. The Safety Benefit ($S_i$) for each Case i was quantified relative to the worst-performing baseline (Z3), considering both severe and moderate hazard zone lengths, which are defined as:

$$S_i = \alpha \left(L_{s,\max} - L_{s,i}\right) + \beta \left(L_{m,\max} - L_{m,i}\right) \tag{3}$$

Where $L_{s,\max}$ and $L_{m,\max}$ are the maximum lengths observed among all Cases. $L_{s,i}$ and $L_{m,i}$ are the lengths of severe and moderate hazard zones for Case i. $\alpha$ and $\beta$ are weighting coefficients (set to $\alpha = 0.8$ and $\beta = 0.2$) to reflect the higher priority of controlling severe hazards.

The Cost $C_i$ was simply represented by the number of zones ($N_i$), as it directly correlates with system complexity and capital expenditure. The overall performance was then evaluated using a Benefit Index ($B_i$), defined as the safety benefit per unit cost (zone):

$$B_i = \frac{S_i}{C_i} \tag{4}$$

The calculated indices are summarized in Table 4.

The results quantitatively demonstrate that the five-zone configuration (Z5) achieves a significantly higher Benefit Index than all other alternatives. This confirms that the Z5 configuration delivers the highest safety return per unit of system complexity, providing a rigorous justification for this optimal recommendation.

Further empirical evidence comes from Fig 9, where temperature distributions across the entire tunnel are compared for all zoning cases (Z3, Z5, Z7, Z9), with explicit marking of the 68°C isotherm.

The temperature analysis further reveals key implications for each design: Case Z3 exhibits the largest thermal risk; Case Z9, with excessive zoning, shows impaired high-temperature removal; Cases Z5 and Z7 both effectively confine high-temperature areas. Given that Case Z5 achieves this safety performance with fewer zones, it represents the optimal balance between hazard control and cost efficiency, thereby confirming its status as the preferred Case.

### Effect of exhaust airflow distribution on smoke extraction

Based on the previous analysis, the low ratio of suction force to inertial force near the fire source hinders effective smoke capture, indicating the requirement for increased airflow in this region. Conversely, farther from the fire source, the horizontal smoke velocity decreases due to wall friction, where excessive airflow may induce the "plug-holing" phenomenon and reduce extraction efficiency [34,35]. Consequently, an imbalanced airflow distribution strategy—with higher extraction rates near the fire source and lower rates at greater distances—is expected to improve smoke control. To test this hypothesis and further optimize the system, different airflow distribution ratios are evaluated under a constant total exhaust volume of 180 m³/s to determine the optimal configuration. Building upon the previous optimized parameters—namely, a smoke screen height of 1.5 m and the tunnel divided into 5 zones—a systematic investigation into the effect of airflow allocation is conducted. The detailed scenario settings are summarized in Table 5.

Fig 10 shows the temporal evolution of CO concentration at the 2-meter safety height at the midpoint of Zone II. As can be seen from Fig 10, CO levels begin to rise at 25 seconds after ignition and stabilize after 150 seconds. It is noteworthy that within the 50–70 second interval, increasing the exhaust airflow allocation near the fire source from 25% to 35%

**Table 4. Comparison of the Benefit Index ($B_i$) for different zoning configurations.**

| Case No. | Safety benefit $S_i$ | Cost $C_i$ | Benefit index $B_i$ |
|---|---|---|---|
| Z3 (3 zones) | 0.0 | 3 | 0.00 |
| Z5 (5 zones) | 58.6 | 5 | 11.72 |
| Z7 (7 zones) | 24.2 | 7 | 3.46 |
| Z9 (9 zones) | 20.6 | 9 | 2.29 |

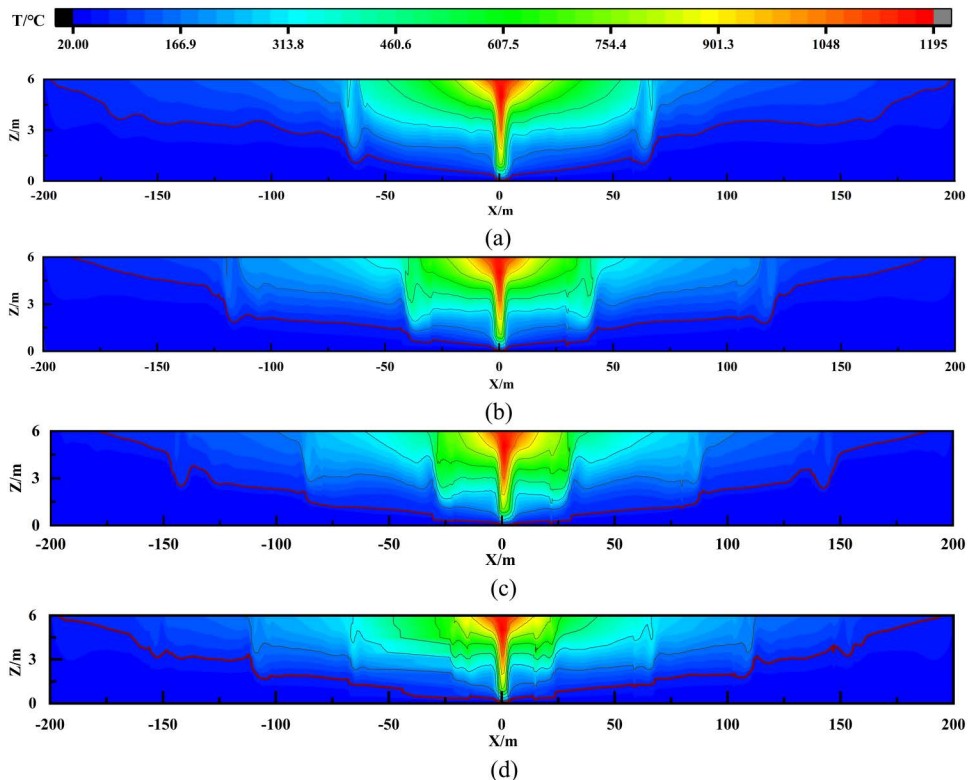

**Fig 9. Smoke temperature distribution in central surface with varying numbers of modular zones: (a) 3 zones; (b) 5 zones; (c) 7 zones; (d) 9 zones.**

**Table 5. Parametric study on airflow distribution.**

| Case No. | Total exhaust volume(m³/s) | Airflow distribution ratio (%) | | | | h (m) | d (m) |
|---|---|---|---|---|---|---|---|
| | | I | II | III | IV | | |
| W1 | 180 | 25 | 25 | 25 | 25 | 1.5 | 2 |
| W2 | | 20 | 30 | 30 | 20 | | |
| W3 | | 15 | 35 | 35 | 15 | | |
| W4 | | 10 | 40 | 40 | 10 | | |
| W5 | | 5 | 45 | 45 | 5 | | |

results in a noticeable decrease in CO concentration, whereas further increasing the allocation from 35% to 45% yields a significantly smaller reduction, indicating that the improvement effect tends toward saturation.

A similar trend can be observed in the average CO concentration profile along the entire tunnel (shown in Fig 11): in the region near the fire source (Vents II/III), increasing the airflow ratio from 25% to 35% progressively reduces the CO concentration from 877.65 ppm (W1) to 716.53 ppm (W3). However, when the ratio is further increased to 45%, the improvement is substantially diminished, with the concentration decreasing only from 697.92 ppm to 669.60 ppm—a clearly weakened reduction. Similarly, in the region farther from the fire source, increasing the airflow ratio from 5% to 15% leads to a pronounced decrease in CO concentration, whereas raising it from 15% to 25% results in a markedly smaller reduction.

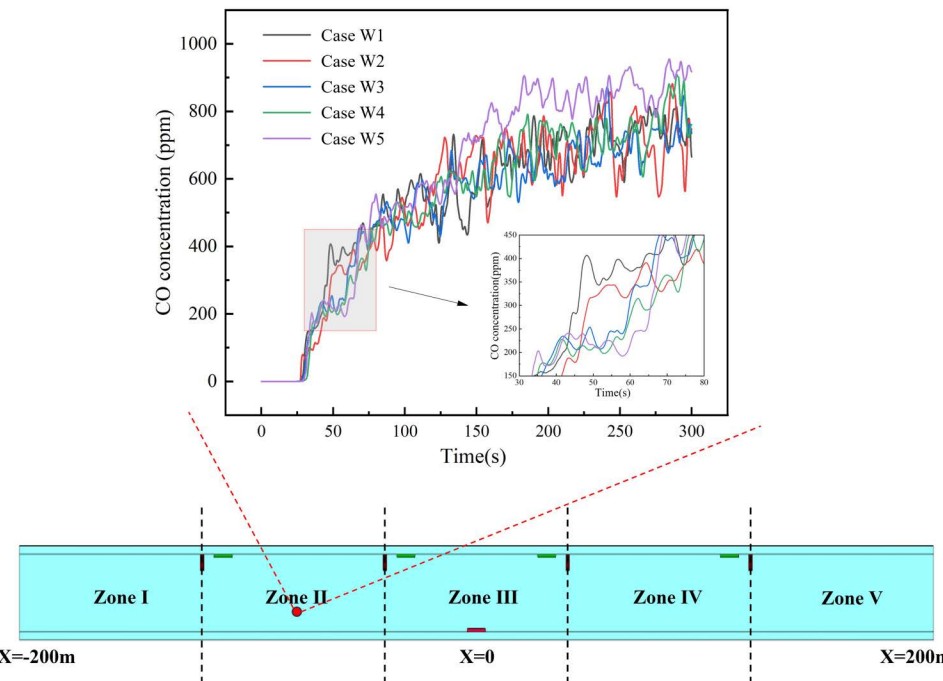

**Fig 10. CO concentration over time at safety height for different scenarios.**

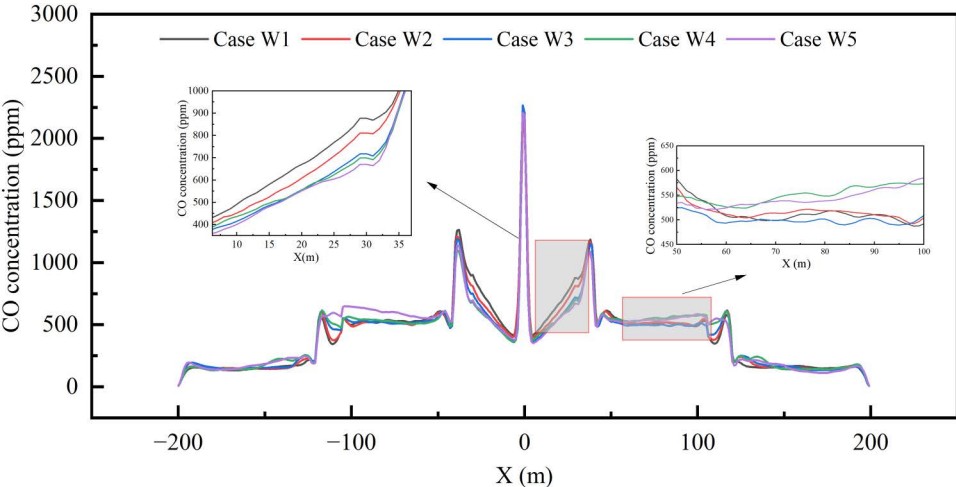

**Fig 11. CO concentration distribution across the entire tunnel at safety height under different scenario.**

The observed patterns collectively suggest that smoke control effectiveness diminishes when the airflow allocation exceeds certain thresholds. The most plausible explanation is the onset of the "plug-holing" phenomenon. Specifically, it is hypothesized that plug-holing is likely to occur at a vent when the airflow allocation exceeds approximately 35% for vents near the fire source, or about 15% for those farther away. This phenomenon would cause the exhaust flow to penetrate the smoke layer, drawing in a portion of fresh air instead of smoke, thereby reducing system efficiency and ultimately manifesting as the observed saturation in CO extraction performance.

This hypothesis is strongly supported by the smoke temperature field. As shown in Fig 12, under the recommended air-flow allocations—30% for the fire-proximate Vent III (Fig 12a) and 5% for the distant Vent IV (Fig 12c)—a smooth and distinct interface is maintained between the high-temperature smoke layer and the underlying air, indicating that the smoke stratification remains intact during extraction. Conversely, when the airflow is increased beyond these thresholds—to 45% for Vent III (Fig 12b) and 20% for Vent IV (Fig 12d)—a prominent triangular low-temperature indentation forms directly beneath each vent. This visual evidence clearly demonstrates that excessive extraction induces ingress of cold air from below, confirming the occurrence of the plug-holing phenomenon.

To more intuitively reveal the flow structure near the exhaust vents and verify the occurrence of the plug-holing phenomenon, the temperature distribution on a horizontal cross-section 0.5 m below the tunnel ceiling (Z = 5.5 m) was further extracted, as shown in Fig 13. In scenarios without plug-holing (Vent III at 30% and Vent IV at 5%), shown in Fig 13a and 13c, the temperature around the vents remains continuous and relatively uniform. This indicates that the vent primarily extracts the upper hot smoke without disrupting the stratified smoke layer. In contrast, under plug-holing conditions (Vent III at 45% and Vent IV at 20%), shown in Fig 13b and 13d, a distinct isolated cold zone forms directly beneath each vent. This clear thermal contrast visually confirms that excessive airflow locally penetrates the smoke layer, drawing fresh cold air directly into the vent and thereby reducing exhaust efficiency.

## Conclusions

A series of numerical simulations led to the following key findings and design principles for optimizing modular smoke extraction with smoke screens in road tunnels:

(1)  The efficiency of the system is dominantly controlled by the smoke screen height, with a critical threshold identified at 1.5 m (25% of the tunnel height). Beyond this height, significant flow separation occurs, creating an upstream recirculation zone that acts as a "smoke reservoir" and drastically improves capture efficiency. In contrast, the

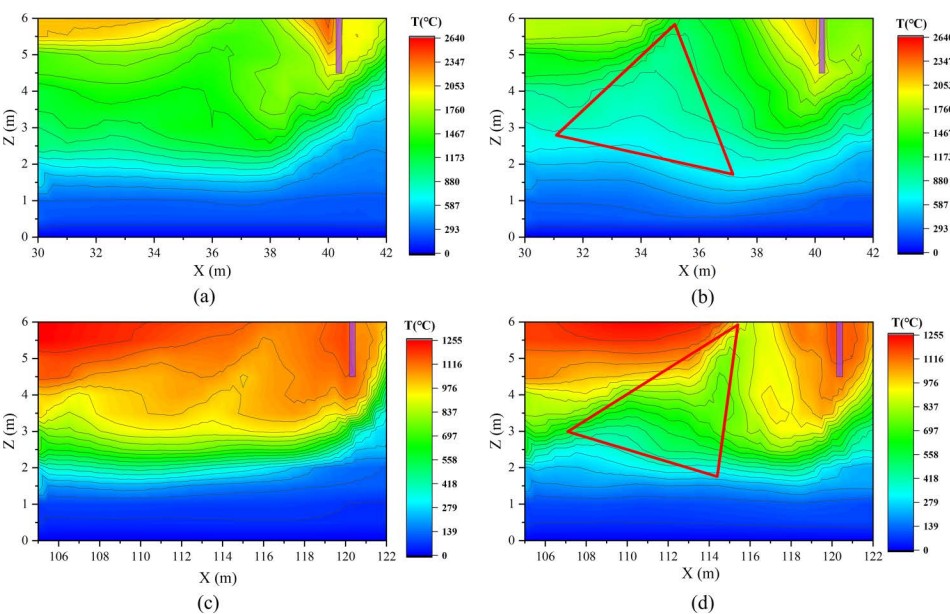

**Fig 12. Temperature distributions on the central longitudinal plane under varying vent and airflow ratios: (a) Vent III at 30%; (b) Vent III at 45%; (c) Vent IV at 5%; (d) Vent IV at 20%.**

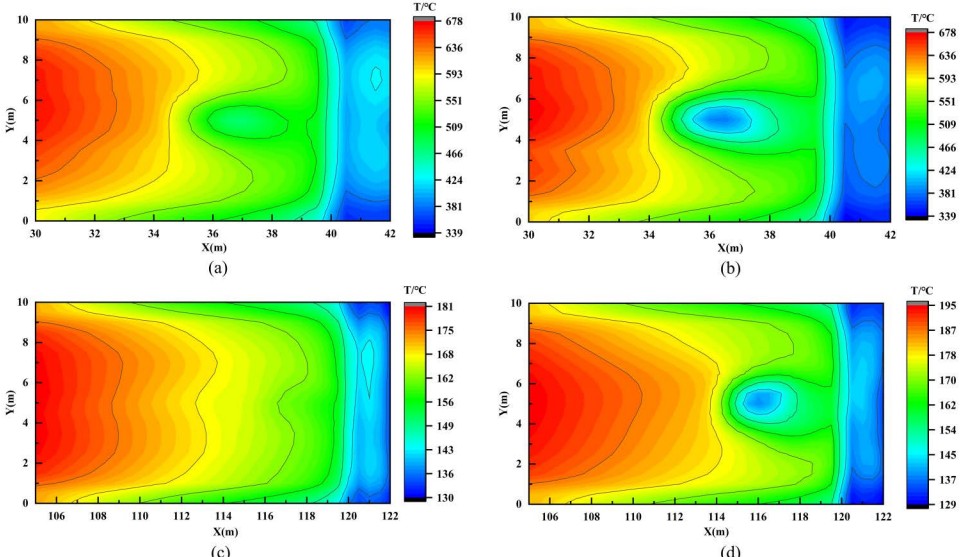

**Fig 13. Temperature distributions on the horizontal plane at Z = 5.5 m: (a) Vent III at 30%; (b) Vent III at 45%; (c) Vent IV at 5%; (d) Vent IV at 20%.**

vent-screen distance (1–3 m) proved to be a secondary factor within the studied range, a finding attributed to the robust suction field generated by the mechanical exhaust system, differentiating it from buoyancy-driven natural ventilation.

(2) For a 400 m tunnel under a 20 MW fire, partitioning into five modular zones represents the optimal cost-benefit trade-off, effectively shortening the free smoke travel distance while maintaining adequate suction power at each vent to contain both severe and moderate hazard zones.

(3) An unbalanced airflow distribution strategy is essential for maximizing performance. Allocating 35% of the total exhaust volume (180 m³/s) to vents near the fire and 15% to distal vents was found to be optimal. This strategy successfully prevents plug-holing while ensuring efficient extraction of CO and heat near the fire source.

## Supporting information

**S1 Data. Data set.**
(ZIP)

## Author contributions

**Conceptualization:** Xiaotao Zhang.

**Data curation:** Kaihua Lu.

**Methodology:** Xiaotao Zhang.

**Software:** Yushi Lu.

**Writing – original draft:** Xiaotao Zhang.

**Writing – review & editing:** Kaihua Lu.

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
