## [Decision Letter · Decision Letter 0]

29 Oct 2025

Dear Dr. Zhang,

Thank you for submitting your manuscript to PLOS ONE. After careful consideration, we feel that it has merit but does not fully meet PLOS ONE’s publication criteria as it currently stands. Therefore, we invite you to submit a revised version of the manuscript that addresses the points raised during the review process.

We look forward to receiving your revised manuscript.

Kind regards,

Gianluca Genovese, Ph.D.

Academic Editor

PLOS ONE

“The works described in this paper are substantially supported by Suqian Sci&Tech Program (Grant No. K202434) and National Natural Science Foundation of China (Grant No. 52376133).”

“The works described in this paper are substantially supported by Suqian Sci&Tech Program (Grant No. K202434) and National Natural Science Foundation of China (Grant No. 52376133)”

“The works described in this paper are substantially supported by Suqian Sci&Tech Program (Grant No. K202434) and National Natural Science Foundation of China (Grant No. 52376133).”

Reviewers' comments:

Reviewer's Responses to Questions

**Comments to the Author**

1. Is the manuscript technically sound, and do the data support the conclusions?

Reviewer #1: Yes

Reviewer #2: Partly

2. Has the statistical analysis been performed appropriately and rigorously?

Reviewer #1: Yes

Reviewer #2: Yes

3. Have the authors made all data underlying the findings in their manuscript fully available?

Reviewer #1: No

Reviewer #2: Yes

4. Is the manuscript presented in an intelligible fashion and written in standard English?

Reviewer #1: Yes

Reviewer #2: No

Reviewer #1: - Please, convert this section " Based on the above analysis, a series of numerical..." to the contribution section

- Authors should include a Flowchart of the problem and the proposed solution,

- Please support the abstract by obtained results

- It could be interesting to summarize the commented literature works in a table to have a clear comparison between all. This could also help precisely formulating the contribution of the paper with respect to previous works

- Add a comparison between your approach and findings against the recent literature approaches and findings.

Reviewer #2: The paper needs proper revision.

1. The CFD modeling section lacks validation of the simulation setup with experimental or previously published benchmark results.

2. The boundary conditions are oversimplified, both tunnel ends are treated as open boundaries, but real tunnels often have semi-enclosed conditions or ventilation shafts.

3. The fire source description (n-heptane pool, 20 MW) is not supported by experimental verification or heat release rate time curve; assuming a steady HRR neglects realistic transient behavior.

4. The mesh independence study is only justified by the D* criterion but not actually demonstrated through sensitivity analysis. The accuracy of results at 0.25-0.5 m mesh remains questionable.

5. The study does not include any uncertainty analysis or sensitivity study on turbulence model parameters???

6. The governing equations and numerical schemes of FDS are briefly cited but not adequately described or justified for the chosen grid scale and fire size.

7. The choice of CO₂ as a tracer for smoke extraction efficiency may not accurately represent smoke behavior; direct smoke or soot mass fraction analysis would be more relevant.

8. The assumption that CO concentration is a surrogate for toxicity and evacuation safety is too simplified; combined effects of CO, temperature, and visibility should be considered.

9. The study claims an optimal smoke screen height of 1.5 m, but this conclusion appears case-specific. No scaling or generalization method is discussed for tunnels with different geometries or fire sizes.

10. The selection of five modular zones as optimal lacks a quantitative cost-benefit analysis or an objective optimization framework; it relies solely on limited simulation scenarios.

11. The influence of longitudinal air movement, natural draught, and vehicle-induced flow has been neglected?? ??

12. The analysis of the plug-holing phenomenon is largely qualitative; no flow field visualization or mass balance is provided to substantiate the mechanism.

13. The smoke extraction efficiency definition based on CO₂ flux is not fully consistent with standard approaches, and the formula needs clearer explanation and justification.

14. Figures (especially Figs. 2–8) lack proper axis labels, quantitative scales, and legends.

15. The study ignores the impact of varying tunnel slope, which in practice influences smoke stratification and backlayering.

16. The discussion section merely restates the results without deeper physical interpretation or comparison with previous numerical or experimental studies.

17. English expression requires improvement; there are frequent grammatical errors and long sentences that reduce readability.

**Do you want your identity to be public for this peer review?** For information about this choice, including consent withdrawal, please see our Privacy Policy

Reviewer #1: No

Reviewer #2: No

---

## [Author Response · Author response to Decision Letter 1]

14 Dec 2025

Dear Editor,

We sincerely thank you for handling the revision of our manuscript entitled "Study on modular smoke extraction with solid screen in urban road tunnel fires" (Manuscript ID: [PONE-D-25-37621]) and for forwarding the highly constructive and detailed comments from the two reviewers. We are truly grateful to the reviewers for their valuable time and effort. Their insightful and professional comments have been instrumental in significantly enhancing the depth of this research and the clarity of its presentation.

We have given the most serious consideration to every comment and have provided comprehensive, point-by-point responses and revisions in the revised manuscript. The scope of these revisions spans several critical aspects, including clarifying the study's core contributions, strengthening the validation of the numerical model, providing a deeper physical interpretation of the key findings, as well as optimizing figures and refining the language.

We are confident that this thorough revision has substantially improved the scientific quality and academic value of the manuscript. Attached please find our detailed point-by-point response letter and the revised manuscript with all changes highlighted.

We once again extend our gratitude to the editorial office and the reviewers for their invaluable guidance. We respectfully request your consideration of the revised manuscript for the next stage of the publication process.

Our specific, point-by-point responses to each reviewer's comments are detailed below:

Reviewer #1

We thank Reviewer #1 for the positive evaluation and the constructive suggestions, which have helped us improve the manuscript. Our point-by-point responses are provided below.

1. Please, convert this section " Based on the above analysis, a series of numerical..." to the contribution section.

Thank you for the constructive feedback. We have carefully revised the manuscript according to your suggestions. Specifically, in response to the comment regarding the contribution statement, we have converted the final paragraph of the Introduction section into a clear contribution summary and this change has been tracked and shown in the revised manuscript using the “Track Changes” mode in Microsoft Word..

The original text concluding the Introduction has been replaced with the following paragraph, which now explicitly states the study's objectives and primary contributions in a concise and formal manner:

"Building upon existing research, this study undertakes a systematic numerical investigation using high-fidelity Computational Fluid Dynamics (CFD) to evaluate a novel modular smoke extraction system with solid screens in a full-scale urban road tunnel. The primary contributions are threefold: (1) establishing a parametric framework to quantify the combined effects of key design variables on global extraction efficiency; (2) identifying optimal design thresholds that balance performance with practical cost; and (3) proposing specific engineering guidelines, including an unbalanced airflow strategy, for system optimization. These outcomes provide direct, actionable insights for the design of advanced transverse ventilation systems in urban tunnels."

We believe this revision strengthens the manuscript by clearly framing its value at the outset. We are grateful for the guidance provided and hope the revised text meets the journal's standards.

2. Authors should include a Flowchart of the problem and the proposed solution

We sincerely thank the reviewer for this valuable and constructive suggestion. Following your recommendation, we have added a clear and concise Flowchart in the revised manuscript to visually outline the research framework. The Flowchart is presented in Fig.1

Fig.1. Flowchart of the research framework

The flowchart systematically illustrates the logical flow of this study:

Research Objective: To achieve global optimization of smoke extraction in urban road tunnels.

Model Setup: The establishment of the CFD model, including tunnel geometry and the configuration of the modular exhaust system with solid screens.

Performance Evaluation Metrics: The key indicators (e.g., extraction efficiency, CO concentration) used to assess system performance.

Optimal Design Parameters: The primary outcomes, identifying the optimal values for critical design variables.

This visual summary enhances the readability and structural clarity of the paper, allowing readers to quickly grasp the problem addressed, the methodology employed, and the solutions derived. We appreciate this insightful suggestion to improve our manuscript's presentation.

3. Please support the abstract by obtained results.

We sincerely thank you for this valuable suggestion. We agree that strengthening the abstract with more concrete findings enhances the clarity and impact of our work.

Accordingly, we have thoroughly revised the abstract to better integrate and highlight the key results obtained from our numerical simulations. The revised abstract now more explicitly presents the specific design thresholds and optimal strategies identified in the study, thereby providing stronger support for the conclusions. The updated text is as follows:

“A series of numerical simulations were conducted to evaluate and optimize a modular smoke extraction system integrated with solid screens for urban road tunnel fires. The study aimed to identify key design parameters and propose optimal strategies for global smoke management. The results show that smoke screen height is a critical factor, with a threshold of 1.5 m (25% of tunnel height) significantly enhancing extraction efficiency and reducing high-temperature zones, while vent-screen distance has minimal impact. Dividing the tunnel into five modular zones achieves the optimal balance between hazard control and system cost. Furthermore, an unbalanced airflow distribution strategy—allocating 35% of the total exhaust volume (180 m³/s) to proximate vents and 15% to distal vents—proves most effective in preventing plug-holing and maximizing CO extraction efficiency. These findings provide specific design thresholds and actionable strategies for the optimization of transverse smoke extraction systems in urban road tunnels.”

This revised version has been updated in the manuscript. We believe these modifications have significantly strengthened the abstract. We are grateful for your insightful comment, which has helped improve the quality of our paper.

4. It could be interesting to summarize the commented literature works in a table to have a clear comparison between all. This could also help precisely formulating the contribution of the paper with respect to previous works.

We sincerely thank the reviewer for the valuable comment. The suggestion to summarize the commented literature in a table for clearer comparison and to help formulate the paper's contribution is very helpful.

Following this suggestion, we have added a summary table, as shown in Table 1. to the Introduction section of the revised manuscript.

Table 1. Summary of literature on mechanical smoke control in tunnel fires.

Ref. Method Core Research Focus

[1] Review Overview of tunnel fire safety design, including smoke control.

[2] Review Foundational small-scale duct experiments on fire spread and ventilation.

[3] Review Fire hazards and mitigation strategies in transportation infrastructures.

[4] Statistical Analysis Analysis of tunnel fire incident statistics (2012-2023).

[5] Experiment & Theory Smoke control strategy and design criterion for "complete smoke extraction".

[6] Review Comparison of longitudinal and transverse ventilation systems for different tunnel types.

[7] Experiment Comparison of longitudinal vs. transverse ventilation effectiveness in a UTLT.

[8] Experiment & Simulation Smoke stratification length under longitudinal ventilation.

[9] Theory & Simulation Quantitative evaluation and optimization of exhaust under lateral centralized mode.

[10] Simulation/Experiment Influence of vent number/layout on full transverse exhaust performance.

[11] Simulation Smoke control with different exhaust patterns and air supply in ultra-wide tunnels.

[12] Simulation Effect of lateral extraction on transverse temperature distribution under ceiling.

[13] Simulation Mechanical smoke extraction efficiency of multiple lateral vents in an immersed tunnel.

[14] Experiment Impact of transverse ventilation on smoke spread in an urban tunnel.

[15] Theory & Simulation Heat and smoke exhaust performance under lateral centralized mode.

[16] Simulation Compartmentalization and smoke control using Water Screens (WS).

[17] Simulation Optimal placement and height of solid curtains near exhaust vents.

[18] Experiment Damper shape and position for smoke confinement under transverse ventilation.

[19] Simulation A novel inclined smoke screen for lateral extraction in an immersed tunnel.

[20] Simulation Solid screen enhancement for shaft extraction in urban road tunnels.

This table directly addresses your point by providing a clear overview of the key literature discussed, which helps to contextualize and precisely frame the contributions of our present work.

Thank you again for this constructive feedback.

5. Add a comparison between your approach and findings against the recent literature approaches and findings.

We sincerely thank you for the valuable suggestion. We fully agree that deepening the physical interpretation of our findings and placing them in a broader research context greatly strengthens the discussion.

Following your guidance, we have thoroughly revised the Discussion section to substantially enhance the theoretical explanation of our key results. The main improvements are reflected in the following two aspects:

Enhanced Comparison and Mechanistic Analysis: We have not only added a detailed comparison of our results with previous studies but, more importantly, elucidated the fundamental physical mechanisms (forced pressure difference vs. thermal buoyancy) behind the differing sensitivities of key parameters in mechanical and natural ventilation systems. This places our findings within a clearer framework of disciplinary understanding.

Deepened Interpretation of Physical Mechanisms: For the threshold effect of "smoke screen height," we moved beyond phenomenological description to provide an in-depth explanation from the perspectives of fluid mechanics mechanisms such as "flow separation," "formation of a recirculation vortex zone," and "dynamic smoke reservoir." This clarifies the essential reason for the performance surge and engages in a comparative and mechanistic discussion with existing empirical guidelines.

We have revised the Discussion section based on the above principles. Some of the key modifications are presented below:

Original Text: Fig 5 further reveals that increasing the vent-screen distance (d) from 1 m to 3 m has negligible impact on exhaust efficiency, indicating d is a secondary factor within this range.

Revised Text: Fig 5 further shows that increasing the vent-screen distance (d) from 1 m to 3 m has a negligible impact on smoke exhaust efficiency, indicating that d is a secondary factor within this range. This finding contrasts with that of Zhang et al. (2022) for a natural ventilation system. In their study, exhaust efficiency was highly sensitive to shaft-screen distance over a short range (0–2.1 m). This discrepancy likely stems from the different driving mechanisms involved: natural ventilation relies primarily on thermal buoyancy, where even minor distance variations can alter the natural convergence path of hot smoke toward the shaft. In contrast, the mechanical transverse exhaust system employed here utilizes fan-induced forced pressure differences, which create an effective suction field across a wider region behind the screen (1–3 m). Consequently, capture efficiency is less sensitive to precise vent-screen spacing within this range.

These revisions aim to make the discussion more insightful and universally significant. Further corresponding refinements have also been made throughout the revised manuscript.

Once again, we are grateful for your insightful comment, which has significantly improved the quality of our paper.

Reviewer #2

We are grateful to Reviewer #2 for the thorough review and the insightful comments, which have significantly strengthened the paper. Our responses follow.

1. The CFD modeling section lacks validation of the simulation setup with experimental or previously published benchmark results.

Thank you for your critical comment regarding the need for validation of the CFD model setup. We fully agree on the importance of validation for ensuring the reliability of simulation results. In response to your suggestion, we have substantially strengthened the validation section in the revised manuscript. Our validation efforts primarily include the following two aspects:

(1) Determination of Model Parameters Based on Established Research

The key parameters for the Large Eddy Simulation (LES), specifically the subgrid-scale model coefficients (Smagorinsky constant Cs=0.18, turbulent Prandtl number Pr=0.5, and turbulent Schmidt number Sc=0.5) were rigorously selected based on authoritative studies concerning buoyancy-driven flows in enclosed spaces. A detailed theoretical justification and literature support for this parameter selection have been comprehensively addressed in our response to Reviewer 2’s Comment #5. This ensures the physical soundness and reliability of our turbulence modeling approach.

(2) Quantitative Validation Against Published Experimental Temperature Data:

To further validate our overall simulation setup, we conducted a dedicated benchmark simulation. We numerically reconstructed the experimental scenario reported by Luo et al. (2013) in Safety Science, which investigated smoke confinement and exhaust efficiency using a modified Opposite Double-Jet Air Curtain. For quantitative comparison, thermocouples were placed in the simulation at locations strictly corresponding to the measurement points specified in the original experimental setup, and temperature data were extracted. The comparative results are presented in Fig 1.

Fig 1. Model validation against experimental temperature data.

From the Fig 1, the comparison shows good agreement between our numerical results and the experimental data, confirming the reliability of our model in predicting the thermodynamic behavior of smoke. A new subsection detailing this benchmark validation has been added to the “Methods” chapter (Section “Model construction”) in the revised manuscript, along with a comparative figure. We believe these additions adequately address your concern and firmly establish the credibility of our simulation methodology.

Thank you again for prompting this important improvement to our work.

2. The boundary conditions are oversimplified, both tunnel ends are treated as open boundaries, but real tunnels often have semi-enclosed conditions or ventilation shafts.

Thank you for your insightful comment regarding the simplification of boundary conditions in our model. We appreciate your emphasis on the complexity of real tunnel environments, such as semi-enclosed portals or ventilation shafts.

In response to your concern, we would like to clarify the rationale behind our modeling choice from the following perspectives:

(1) Focus on Preliminary Feasibility and Mechanism Isolation

Our current study represents a preliminary investigation into the feasibility of a novel modular smoke control concept. At this initial stage, our primary objective is to explore whether this modular approach can function as a viable smoke management strategy, focusing on key design variables such as solid screen height, zone partitioning, and exhaust airflow distribution. The use of open boundaries at both tunnel ends establishes a benchmark, controlled flow environment for this feasibility assessment. This simplified configuration intentionally isolates the proposed system from external complexities such as portal effects or coupled shaft ventilation, allowing for a clearer preliminary evaluation of whether the modular concept demonstrates fundamental viability under controlled conditions.

(2) Common Practice in Founda

---

## [Decision Letter · Decision Letter 1]

26 Jan 2026

Study on modular smoke extraction with solid screen in urban road tunnel fires

PONE-D-25-37621R1

Dear Dr. Zhang,

We’re pleased to inform you that your manuscript has been judged scientifically suitable for publication and will be formally accepted for publication once it meets all outstanding technical requirements.

Kind regards,

Gianluca Genovese, Ph.D.

Academic Editor

PLOS One

Reviewers' comments:

Reviewer's Responses to Questions

**Comments to the Author**

Reviewer #1: (No Response)

Reviewer #2: All comments have been addressed

2. Is the manuscript technically sound, and do the data support the conclusions?

Reviewer #1: (No Response)

Reviewer #2: Yes

3. Has the statistical analysis been performed appropriately and rigorously?

Reviewer #1: (No Response)

Reviewer #2: Yes

4. Have the authors made all data underlying the findings in their manuscript fully available?

Reviewer #1: (No Response)

Reviewer #2: Yes

5. Is the manuscript presented in an intelligible fashion and written in standard English?

Reviewer #1: (No Response)

Reviewer #2: Yes

Reviewer #1: no further comments

no further comments

no further comments

no further comments

no further comments

Reviewer #2: The authors have improved the manuscript significantly and is therefore recommended for the acceptance.

**Do you want your identity to be public for this peer review?** For information about this choice, including consent withdrawal, please see our Privacy Policy

Reviewer #1: No

Reviewer #2: No

---

## [Editor Report · Acceptance letter]

PONE-D-25-37621R1

PLOS One

Dear Dr. Zhang,

I'm pleased to inform you that your manuscript has been deemed suitable for publication in PLOS One. Congratulations! Your manuscript is now being handed over to our production team.

Kind regards,

on behalf of

Dr. Gianluca Genovese

Academic Editor

PLOS One